# Best Practice PD-L1 Staining and Interpretation in Gastric Cancer Using PD-L1 IHC PharmDx 22C3 and PD-L1 IHC PharmDx 28-8 Assays, with Reference to Common Issues and Solutions

**DOI:** 10.3390/biomedicines13112824

**Published:** 2025-11-19

**Authors:** Soomin Ahn, Inwoo Hwang, Yuyeon Kim, Somin Lee, Yunjoo Cho, So Young Kang, Deok Geun Kim, Jeeyun Lee, Kyoung-Mee Kim

**Affiliations:** 1Department of Pathology and Translational Genomics, Samsung Medical Center, Sungkyunkwan University School of Medicine, Seoul 06351, Republic of Korea; suminy317@gmail.com (S.A.); inwoo87.hwang@samsung.com (I.H.);; 2Advanced Technology Research Center for Diagnostics (ATRCD), Samsung Precision Genome Medicine Institute (SPGM), Samsung Medical Center, Seoul 06351, Republic of Korea; yuyeon12.kim@sbri.co.kr (Y.K.);; 3Department of Medicine, Division of Hematology-Oncology, Samsung Medical Center, Sungkyunkwan University School of Medicine, Seoul 06351, Republic of Korea

**Keywords:** gastric, cancer, PD-L1, biomarker, staining, interpretation

## Abstract

Programmed death-ligand 1 (PD-L1) biomarker testing in gastric cancer is required to identify patients suitable for immunotherapy. However, the PD-L1 testing landscape is complex, with various PD-L1 tests available and multiple algorithms that combine tumor and immune cell staining. To provide guidance on the best practices for PD-L1 testing in gastric cancer, we reviewed the literature and incorporated our extensive experience using the PD-L1 IHC PharmDx 22C3 and 28-8 assays and scoring with the combined positive score (CPS) algorithm. This review summarizes inter-reader agreement and PD-L1 assay concordance studies in gastric cancer, highlights practical challenges and pitfalls encountered in our own laboratory, and proposes solutions to address them. Accurate and consistent interpretation of PD-L1 CPS in gastric cancer is challenging, but can be improved with training, experience, and close attention to interpretation guidelines. Techniques are available that can optimize the automated staining of PharmDx PD-L1 assays using the Autostainer Link 48 to ensure consistent staining performance. The PD-L1 IHC PharmDx 22C3 and PD-L1 IHC PharmDx 28-8 assays show high concordance when used according to manufacturers’ guidelines.

## 1. Introduction

In recent years, the first-line treatment landscape for advanced/metastatic gastric cancer has become increasingly complex. Depending on biomarker status, patients can be prescribed HER2-directed therapeutics and/or immunotherapies added onto a backbone of fluoropyrimidine plus platinum-based chemotherapy [1,2,3]. Patients with HER2-positive tumors may receive pembrolizumab (an anti-PD-1 antibody therapy) in addition to trastuzumab and chemotherapy if their tumors express Programmed death-ligand 1 (PD-L1) above a cutoff of combined positive score (CPS) ≥1, as determined using the PD-L1 IHC PharmDx 22C3 immunohistochemistry (IHC) assay. For patients with HER2-negative tumors, a different anti-PD-1 therapy, nivolumab, is indicated for patients with PD-L1 CPS ≥5 using the PD-L1 IHC PharmDx 28-8 assay. In the US and across Asia, pembrolizumab is approved for treatment of patients with HER2-negative tumors regardless of PD-L1 levels, but in the European Union (EU), pembrolizumab is only indicated for patients with tumor PD-L1 CPS ≥ 1 (Figure 1), and Asian guidelines explicitly state that the benefits of anti-PD-1 antibody plus chemotherapy are more pronounced in the PD-L1-positive group [2,3,4,5]. Tislelizumab has also been recently approved by the U.S. Food and Drug Administration (FDA) in PD-L1 positive, HER2-negative gastric cancer based on findings in the Rationale-305 study [6,7].

Each approved PD-1 therapeutic is associated with a different PD-L1 IHC assay, based on clinical utility data obtained in pivotal clinical trials. Pembrolizumab is associated with the PD-L1 IHC PharmDx 22C3 assay (hereafter referred to as 22C3), whereas nivolumab is associated with the PD-L1 IHC PharmDx 28-8 assay (hereafter referred to as 28-8). A key question for clinical laboratories is whether the two assays perform in the same way and can be used interchangeably. Even minor discrepancies between assays at clinical cutoff points can result in differing treatment eligibility decisions. Assay discordance at clinically relevant CPS thresholds (e.g., CPS ≥1 and ≥5) remains a significant challenge in PD-L1 testing. Even minor discrepancies between assays can lead to divergent treatment eligibility decisions, potentially affecting patient access to immune checkpoint inhibitors. For example, a patient assessed as CPS 4 using the 28-8 assay would not meet the eligibility criteria for pembrolizumab, whereas the same sample evaluated with the 22C3 assay yielding CPS 5 would qualify for treatment. Such inconsistencies may result in clinical misclassification, underscoring the need for standardized interpretation criteria and harmonization across assays. To mitigate these risks, laboratories should adhere to validated companion diagnostic protocols and ensure rigorous quality control throughout the testing process. Greater alignment among pathologists, manufacturers, and regulatory bodies will be essential to improve assay comparability and optimize patient selection.

PD-L1 IHC assay concordance in gastric cancer is an important question for several reasons. Firstly, running multiple PD-L1 tests in one laboratory requires the separate validation of each assay, which is time-consuming and expensive. For most laboratories, the validation of multiple assays is not cost-effective. Also, having more than one test for the same biomarker may cause confusion and limit the ability to save costs due to scale. Secondly, limited sample may be available for some gastric cancer patients, especially where only endoscopic or peritoneal biopsies are available, meaning there may be insufficient sample to run both the 22C3 and 28-8 assays.

The CPS algorithm used for the determination of PD-L1 status in gastric cancer includes both tumor cells (TCs) and immune cells (ICs), and is challenging to score reproducibly [8]. Outside of Asia, gastric cancer is relatively uncommon, making up less than 2% of diagnosed cancers in the US [9], resulting in substantial variations in pathologist experience of such cancers depending on location. Levels of gastric cancer in South Korea are among the highest in the world, with an incidence of 17.6 per 100,000 in females and 39.7 in males [10]. In our facility, we use both 22C3 and 28-8 assays routinely to stain and score gastric cancer samples. We have identified several potential pitfalls and challenges with the use and scoring of these PD-L1 assays, including some specific to gastric cancer. In this paper, we share our experiences, together with findings from the literature, to provide practical guidance on how to improve the consistency and reliability of PD-L1 assessment in gastric cancer samples.

## 2. PD-L1 Staining Using 22C3 and 28-8 Assays—Common Issues and Solutions

When used as indicated, 22C3 and 28-8 assays are both performed using the Autostainer Link 48 (ASL48; Dako, Carpinteria, CA, USA) platform for automated staining. The staining process includes an integrated pre-treatment module that combines deparaffinization, rehydration, and target retrieval in one step. This occurs via the PT link, a specialized instrument used for preparing tissue samples before IHC, and automates the pre-treatment process. In the PT Link, heat-induced antigen retrieval is typically performed within a temperature range of 65 °C to 97 °C. This can cause the plastic racks to deform slightly due to high temperatures. When transferred to the AutoLink 48 stainer, they may tilt and become loose, which increases the risk of dry artifacts and heterogeneous staining. To address this, different racks can be used for the PT Link and the stainer. Alternatively, regular inspection of racks for deformation should be performed, and any deformed racks should be replaced.

Although the Autostainer Link 48 is designed as a closed system, the occasional drying of tissue sections can occur for reasons that are not fully understood. This can be prevented by placing wet paper towels underneath the staining area (Figure 2). However, this is a local recommendation from our laboratory and is not endorsed by the manufacturer. It should be applied with caution and in accordance with appropriate quality assurance protocols.

Compared with the 22C3 assay, the 28-8 assay often shows non-specific staining. As previously reported by our group, non-specific staining was found in 29 (52.7%) cases out of 55 gastric adenocarcinoma samples [11]. Non-specific staining in muscular or glandular tissue was identified in 24 (43.6%) cases, and non-specific cytoplasmic staining in tumor cells was identified in 5 (9.1%) cases [11]. In contrast, matched 22C3-stained slides did not reveal non-specific staining [11]. Examples of non-specific staining with the 28-8 assay are shown in Figure 3A–D.

Non-specific staining can be caused by various factors, such as antibodies, pre-analytic fixation, and processing of the specimen, although sometimes the cause is not known. Antibodies used in the 22C3 and 28-8 assays target the extracellular domain of the protein, and so the epitope is more at risk of denaturation via suboptimal processing than antibodies raised against internal epitopes [12,13]. To minimize issues relating to pre-analytical factors, we recommend using fresh-cut slides for PD-L1 assessment. Slides should be stained within 2 weeks and stored in a refrigerator rather than at room temperature [14]. When using archival formalin-fixed paraffin-embedded (FFPE) blocks, it is recommended to select the most recent specimen whenever possible to minimize antigen degradation. According to one reference guideline for PD-L1 testing in non-small cell lung cancer, FFPE blocks should preferably be used within approximately three years [15]. The use of high-quality formalin for fixation is also important. The checklist for a laboratory is provided in Appendix A. Non-specific staining is emphasized in the interpretation manual of the 28-8 PharmDx assay [14]. Increased awareness of this issue among the pathology community may increase consistency in the interpretation of the 28-8 PharmDx assay.

## 3. Scoring CPS

The definition of CPS is shown in Figure 4. The calculation of CPS is not straightforward, as many factors need to be considered. A specific requirement for gastric cancer PD-L1 scoring is that areas of ulcer and gastritis should be excluded. Within the CPS algorithm, tumor cells with membrane staining are counted, whereas membrane and/or cytoplasmic staining is included for immune cells, of which only lymphocytes and macrophages are counted. Discriminating between lymphocytes and granulocytes/plasma cells in IHC-stained slides is challenging, and if granulocytic/plasmacytic infiltrations are suspected, comparing IHC with Hematoxylin and Eosin (H&E) slides is necessary to identify neutrophils and plasma cells, which should then be excluded from the IC component of the CPS. As a further complexity, only ICs directly associated with the tumor response should be scored [14].

Exclusions and H&E Confirmation (adapted from PD-L1 IHC 28-8 pharmDx Manual for Gastric Cancer)

Exclude from CPS:○Tumor cells with cytoplasmic-only staining;○PD-L1–positive benign glands or metaplastic epithelium;○Fibroblasts, smooth muscle, endothelial cells;○Neutrophils, eosinophils, plasma cells;○Necrotic or mucinous debris.Confirm with H&E:○Verify PD-L1–positive cells correspond to viable tumor or mononuclear immune cells;○Reassess areas with ambiguous staining to avoid inclusion of benign or stromal elements.

An additional challenge in gastric cancer is that it is usually diagnosed by small endoscopic biopsies, and spatial heterogeneity has been observed. To address heterogeneity, where a sample includes both PD-L1-positive and -negative fragments, the final CPS was calculated as the mean value across evaluable fragments. It has been shown that sampling at least four biopsies with a total area of about 4.5 mm^2^ and averaging the CPS can give PD-L1 results similar to resection specimens [16]. Ye et al. reported that PD-L1 expression in tissue microarray samples showed varying degrees of concordance with corresponding surgical specimens, and recommended obtaining at least five biopsies to ensure the accurate evaluation of PD-L1 status [17,18]. This finding underscores the importance of accounting for spatial heterogeneity when assessing biomarker expression in endoscopic biopsies. To mitigate sampling bias and improve diagnostic reliability, several strategies can be employed. These include collecting multiple biopsies from different tumor regions, evaluating multiple histologic sections per biopsy, and integrating immunohistochemical results across sections to capture intratumoral variability. Additionally, incorporating digital pathology tools and quantitative image analysis may enhance the consistency and reproducibility of PD-L1 scoring. Future studies should aim to standardize biopsy protocols and validate these approaches to ensure robust biomarker assessment in clinical practice.

## 4. Inter-Reader Agreement for CPS Scoring in Gastric Cancer

The immune cell component of CPS is the most difficult to score, and inter-observer agreement for PD-L1 IC staining is generally worse than for tumor cell scoring across many different tumor types [19,20]. Consequently, inter-observer agreement is generally lower for CPS than for tumor proportion score when the same assays are used [21]. One study identified several factors in samples that show poor concordance between pathologists, including the ambiguous identification of positively staining stromal cells, faint or variable intensity of staining, difficulty in distinguishing membranous from cytoplasmic tumor staining, and cautery or crush artifacts [22]. In the study, inter-observer agreement among 12 US and EU pathologists assessing PD-L1 using CPS was poor, with an intraclass coefficient of 0.45, and a 2 h CPS training session did not significantly improve the inter-observer agreement [22]. Another US study also demonstrated poor inter-pathologist agreement for CPS scoring in gastric cancer [23]. However, a recent study found a high intraclass coefficient (ICC) (0.92–0.96) for CPS scoring using 22C3 and 28-8 assays [24], and in a Korean study, high agreement was seen, with inter-observer variability of 0.89 and 0.88 for the 28-8 and 22C3 assays, respectively [25]. The lower inter-reader agreement observed in Western countries compared to Eastern results may be attributed to the relatively low incidence of gastric cancer and the limited experience with such cases. Therefore, interpreting PD-L1 CPS likely requires a learning curve, and the improved concordance in Western assessments shown in Klempner’s 2024 [24] study objectively supports this notion. We believe that external validation in other geographic and practice settings is necessary to support broader applicability.

The higher agreement in the more recent studies may reflect higher levels of experience. Intensive training has also been shown to improve CPS scoring reproducibility across multiple cancer types, including gastric cancer [26]. However, it should be noted that even amongst experienced pathologists, including those specializing in gastrointestinal pathology, inter-reader variability was observed for samples close to the CPS ≥5 cutoff [8]. In Korea, pathologists actively engaged in discussions and shared their experiences regarding PD-L1 CPS scoring during consensus meetings, which led to improved consistency in inter-reader agreement. The challenges highlighted included staining heterogeneity across sample fragments and non-specific staining with the 28-8 assay.

## 5. Interchangeability of 22C3 and 28-8 Assays in Gastric Cancer

Several studies have assessed the concordance of 22C3 and 28-8 in gastric cancer (Table 1). Studies have used various concordance metrics and reported that concordance may also vary depending on the sample type. Direct comparison across studies should be interpreted with caution because different metrics and sample types were used.

Excellent concordance between 28-8 and 22C3 was observed in studies that used the PharmDx assays, and in Klempner et al. the comparability of assays was further supported by independent digital image analysis. In our experience, the two assays show similar staining (discounting the non-specific staining that is sometimes seen with the 28-8 assay), and examples of highly concordant samples from our laboratory are shown in Figure 5A–D.

Although some studies found poorer agreement between assays, these studies compared laboratory-developed tests (LDTs) rather than the approved companion diagnostic assay. In Yeong et al., 22C3 or 28-8 primary antibody clones were applied to Leica Bondmax with a dilution ratio that was not described, and detected by multiplex IHC/immunofluorescence. As noted in a subsequent letter to the editor, no data was provided to confirm that these LDTs performed in the same way as the respective companion diagnostic assays, and so their results cannot be extrapolated [29]. Furthermore, samples in this study were 9-21 years in age, and the use of older samples has been reported to impact 22C3 antigenicity and, therefore, associated staining [13]. Notably, the reported prevalence of CPS ≥ 1 using the 28-8 assay in the study was 70.3%, significantly higher than any of the other studies using Asian patient samples [11,21,27]. As noted above, the 28-8 assay is prone to non-specific staining patterns in the gastric mucosa, which can impact interpretation.

Recently, Kim et al. reported suboptimal agreement between the 28-8 PharmDx and 22C3 antibody concentrate, with Cohen’s kappa values and Overall Percentage Agreement between the two assays being 78.3% and 0.56 for the CPS 1 cutoff, 81.8% and 0.60 for the CPS 5 cutoff, and 88.8% and 0.66 for the CPS 10 cutoff, respectively [25]. Kim et al. used diluted PD-L1 22C3 antibody (1:100 from Agilent) on the Ventana Benchmark Ultra, rather than the Autostainer Link, which should also be considered an LDT [25]. Although it has been demonstrated that LDTs with good performance can be developed using the 22C3 primary antibody on the Ventana Benchmark, a 1:50 dilution was found to give the closest agreement with the PharmDx assays [30]. Overall, published data supports a high concordance between the approved 28-8 and 22C3 PharmDx assays. When LDTs are used, it is vital to validate performance by direct comparison with the appropriate gold standard assay.

## 6. Digital Pathology and Artificial Intelligence-Assisted PD-L1 Interpretation

Many researchers have identified image analysis algorithms as a promising tool for enhancing the accuracy and reproducibility of PD-L1 scoring by pathologists in various solid tumors [17,31,32]. Kim et al. applied the Aperio IHC membrane image analysis algorithm (ScanScope™, Aperio Technologies, Vista, CA, USA) to generate PD-L1 CPSs for 39 cases of gastric cancer [17,33]. With additional manual annotation and computational input, they demonstrated that image analysis-supported PD-L1 CPSs were concordant with manual scoring by pathologists [17,33]. More recently, artificial intelligence-assisted PD-L1 image analysis algorithms have shown clinical utility as diagnostic aids in other tumors, such as lung cancer [17,34,35]. However, AI-assisted PD-L1 assessment has not yet been explored in gastric cancer, though advancements in this area are anticipated [36].

## 7. Summary

The accurate and consistent interpretation of PD-L1 CPS in gastric cancer is challenging. Intensive training and/or greater experience in gastric cancer assessment are associated with improved inter-pathologist agreement [26]. Despite ongoing efforts, the variability between assays and pathologists remains, particularly at clinically relevant CPS thresholds. To address this, the standardization of interpretation criteria and continuous education are essential. Laboratories should employ validated companion diagnostic assays in accordance with clinical guidelines, ensure proper pre-analytical handling, and correlate PD-L1 staining with H&E morphology to minimize misinterpretation. Continued harmonization among pathologists, assay manufacturers, and regulatory agencies will be critical to improving assay comparability and optimizing patient selection for immune checkpoint inhibitor therapy. There is a need for the continuous education of practicing pathologists on the specific challenges of CPS scoring in gastric cancer, and the development of external quality assurance programs would help drive improvements. The implementation of objective techniques (e.g., digital pathology) may improve standardization.

## Figures and Tables

**Figure 1 biomedicines-13-02824-f001:**
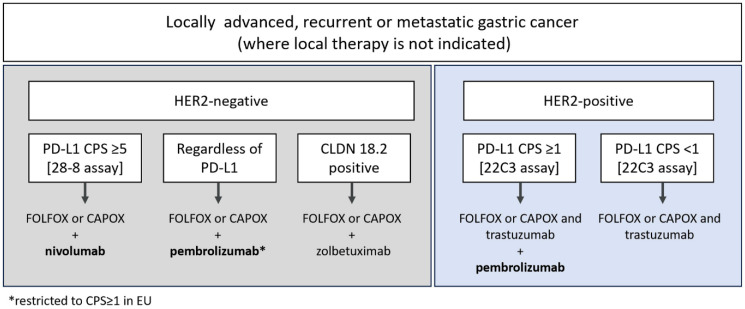
**Therapeutic options in first-line gastric cancer, as indicated in Pan-Asian adapted European Society for Medical Oncology (EMSO) guidelines and National Comprehensive Cancer Network (NCCN), version 5, 2024** [1,2]. CLDN 18.2: Claudin-18.2; PD-L1: Programmed death-ligand-1.

**Figure 2 biomedicines-13-02824-f002:**
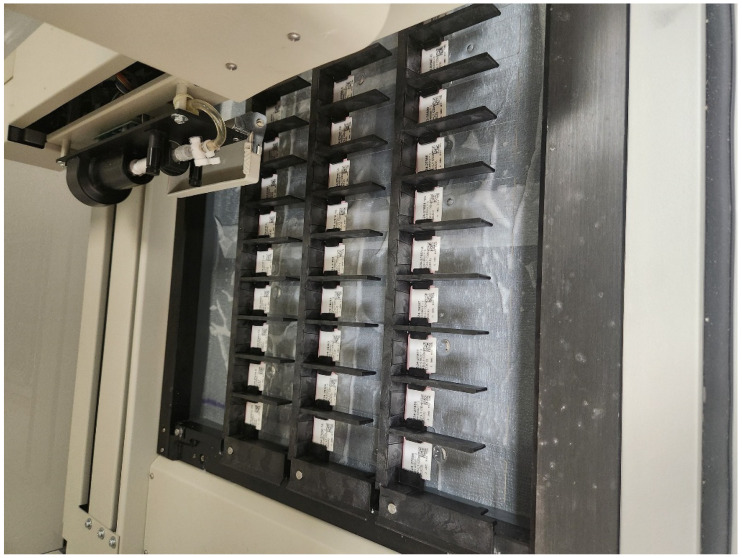
Addressing drying out within Autostainer Link 48 by use of wet paper towels.

**Figure 3 biomedicines-13-02824-f003:**
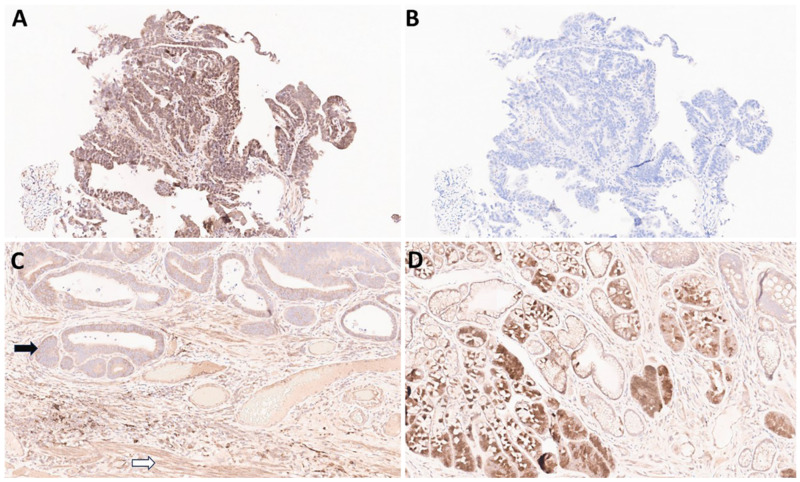
**Examples of non-specific staining of gastric cancer tissues with the PD-L1 IHC PharmDx 28-8 assay.** (**A**): 28-8, 20× magnification. Non-specific staining in tumor and stroma: non-specific cytoplasmic staining without membrane staining in tumor cells should be excluded for CPS calculation. (**B**): 22C3, 20× magnification of the same tissue sample (adjacent section) as in panel (**A**). CPS = 0. (**C**): 28-8, 20× magnification. Non-specific staining in tumor (black arrow) and muscle (white arrow). (**D**): 28-8 20× magnification. Non-specific staining in normal glands.

**Figure 4 biomedicines-13-02824-f004:**
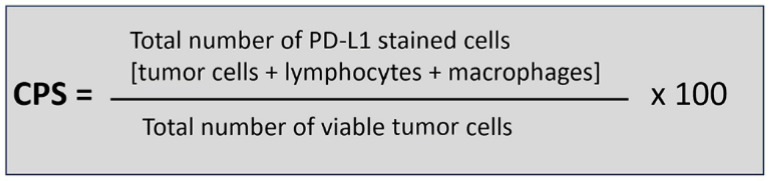
Combined positive score (CPS) algorithm.

**Figure 5 biomedicines-13-02824-f005:**
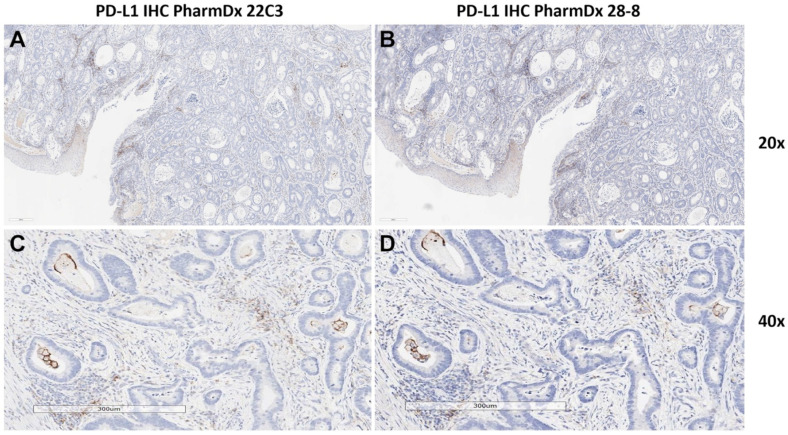
**Examples of gastroesophageal junction cancer samples stained with 22C3 or 28-8, demonstrating similar staining patterns.** (**A**): 22C3, 20× magnification. Focal PD-L1 positivity in immune cells adjacent to carcinoma cells. (**B**): 28-8, 20× magnification. Focal PD-L1 positivity in immune cells adjacent to carcinoma cells, showing staining intensity comparable to that seen with the 22C3 assay. (**C**): 22C3, 40× magnification. Higher magnification of the region shown in (**A**,**D**): 28-8 40× magnification. Higher magnification of the region shown in (**B**).

**Table 1 biomedicines-13-02824-t001:** **Concordance studies of 28-8 and 22C3 assays in gastric cancer.** * OPA data was calculated from Supplemental Figure 2 of [24].

Reference	Assays	n/Sample Type	OPA	Kappa
[27]	28-8 vs. 22C3 (PharmDx assays)	226 TMA	CPS > 1: 90.2% CPS > 5: 97.8% CPS > 10: 97.8%	CPS > 1: 0.735 CPS > 5: 0.881 CPS > 10: 0.837 (Fleiss)
[11]	28-8 vs. 22C3 (PharmDx assays)	55 resections	CPS > 1: 96.4% CPS > 5: not reported CPS > 10: 96.4%	CPS > 1: 0.927 CPS > 5: not reported CPS > 10: 0.899
[28]	28-8 vs. 22C3 (multiplex assay using Leica Bondmax)	362 TMA	CPS > 1: 62.2% CPS > 5: 73.3% CPS > 10: 85.2%	CPS > 1: 0.276 CPS > 5: 598 CPS > 10: 0.818 (Gwet’s)
[25]	28-8 vs. 22C3 (Ventana Benchmark)	143	CPS > 1: 78.3% CPS > 5: 81.8% CPS > 10: 88.8%	CPS > 1: 0.56 CPS > 5: 0.6 CPS > 10: 0.66 (Cohen’s)
[24]	28-8 vs. 22C3 (PharmDx assays)	96 resections	CPS > 1: 85.4% * CPS > 5: 85.4% CPS > 10: 93.7%	Not reported

Abbreviations: OPA, Overall Percentage Agreement; CPS, Combined Positive Score; n, Numbers; TMA, Tissue Microarray.

## Data Availability

The original contributions presented in this study are included in the article/Appendix A. Further inquiries can be directed to the corresponding author.

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
