# Peer review of "Best Practice PD-L1 Staining and Interpretation in Gastric Cancer Using PD-L1 IHC PharmDx 22C3 and PD-L1 IHC PharmDx 28-8 Assays, with Reference to Common Issues and Solutions"

_biomedicines, 2025, doi:10.3390/biomedicines13112824_

Round 1

Reviewer 1 Report

Comments and Suggestions for Authors

Dear authors,
We appreciate the opportunity to review this manuscript, which addresses a topic of great clinical relevance in oncological pathology. However, there are several aspects that merit critical review to strengthen the scientific rigor and clarity of the work:

  1. The manuscript is presented as a review, but includes previously unpublished data (e.g., experience with >50 samples per week). This creates confusion: Is this a narrative review or an observational study?.
  2. Although studies showing high concordance are cited (e.g., Narita et al., Ahn and Kim 2021), studies reporting clinically relevant discordance, especially near cutoff points (CPS ≥1 or ≥5), are also omitted to be thoroughly discussed.
  3. Geographic and Experience Bias
    • It is suggested that the greater experience of Korean pathologists explains the better interobserver agreement, but whether this may limit the generalizability of the conclusions to centers with lower case volumes is not addressed.
  4. Studies using LDTs ​​are criticized (e.g., Yeong et al., Kim H.D. et al.), but no clear criteria are provided for validating LDTs ​​against standard assays.
  5. Although spatial heterogeneity in endoscopic biopsies is mentioned, specific strategies to manage it (e.g., use of multiple biopsies, evaluation of multiple sections) are not discussed.
  6. The role of digital pathology or AI in standardizing PD-L1 assessment is not mentioned, despite recent studies supporting its usefulness.

Recommendations:

a)  Clarify the methodological design and, if applicable, include a “Materials and Methods” section detailing data collection, inclusion criteria, and statistical analysis.

b) Include a balanced discussion of the limitations of interchangeability, especially in contexts of heterogeneous expression or near the cutoff point.

c) Include a discussion on the need for external validation in populations with lower incidence of gastric cancer.

d) Include a brief guide or reference to international standards (e.g., IASLC, CAP) for the validation of LDTs ​​in PD-L1.

e) Include practical recommendations based on recent evidence to improve the representativeness of biopsies.

f) Include a brief section or mention of the future of PD-L1 assessment with digital tools.

Author Response

Dear Reviewer #1

The revised manuscript titled “Best Practice PD-L1 Staining and Interpretation in Gastric Cancer using PD-L1 IHC PharmDx 22C3 and PD-L1 IHC PharmDx 28-8 Assays, with Reference to Common Issues and Solutions” is enclosed for consideration as a review article in Biomedicines. Our responses are detailed below. In the revised manuscript, the changes are highlighted in red.

We appreciate the opportunity to review this manuscript, which addresses a topic of great clinical relevance in oncological pathology. However, there are several aspects that merit critical review to strengthen the scientific rigor and clarity of the work.

Comment 1. The manuscript is presented as a review, but includes previously unpublished data (e.g., experience with >50 samples per week). This creates confusion: Is this a narrative review or an observational study?

Response: We sincerely thank the reviewer for this insightful comment. We acknowledge that the previous version of the manuscript may have caused ambiguity regarding the nature of the study. To clarify, this manuscript is a narrative review grounded in published literature, complemented by the authors’ practical insights from routine diagnostic experience. It is not intended as an observational study. To avoid confusion, all references to unpublished quantitative data have been either removed or appropriately rephrased to reflect experiential context rather than empirical findings (highlighted in red).

Comment 2. Although studies showing high concordance are cited (e.g., Narita et al., Ahn and Kim 2021), studies reporting clinically relevant discordance, especially near cutoff points (CPS ≥1 or ≥5), are also omitted to be thoroughly discussed.

Response: Thank you for your insightful comment. We have added a detailed discussion addressing clinically relevant discordance, particularly near cutoff points (CPS ≥1 or ≥5), in lines 247–250. 

Comment 3. Geographic and Experience Bias • It is suggested that the greater experience of Korean pathologists explains the better interobserver agreement, but whether this may limit the generalizability of the conclusions to centers with lower case volumes is not addressed.

Response: Thank you for this important comment. We acknowledge that most interobserver concordance studies have been conducted in high-volume Asian centers, where pathologists possess substantial experience with PD-L1 scoring in gastric cancer. We now explicitly address the potential limitation this poses to the generalizability of our conclusions, particularly for institutions with lower case volumes. We also emphasize the need for external validation in diverse geographic and practice settings. This clarification has been added to the revised manuscript (line 199-200). 

Comment 4. Studies using LDTs are criticized (e.g., Yeong et al., Kim H.D. et al.), but no clear criteria are provided for validating LDTs against standard assays.

Response: Thank you for your valuable comment. We fully agree that validation of laboratory-developed tests (LDTs) should adhere to internationally recognized standards. However, in routine clinical practice, PD-L1 testing for gastric cancer is predominantly conducted using FDA-approved companion diagnostic assays (e.g., 22C3 PharmDx), which have already undergone rigorous analytical and clinical validation. As such, while LDT validation is important in research contexts, it is generally not a critical concern in daily diagnostic workflows. For this reason, we did not address LDT validation criteria in detail in the current manuscript. 

Comment 5: Although spatial heterogeneity in endoscopic biopsies is mentioned, specific strategies to manage it (e.g., use of multiple biopsies, evaluation of multiple sections) are not discussed.  

Response: We added more discussions in the final manuscript (Ye et al. reported that PD-L1 expression in tissue microarray samples showed varying degrees of concordance with corresponding surgical specimens, and recommended obtaining at least five biopsies to ensure accurate evaluation of PD-L1 status [17,18]. This finding underscores the importance of accounting for spatial heterogeneity when assessing biomarker expression in endoscopic biopsies. To mitigate sampling bias and improve diagnostic reliability, several strategies can be employed. These include collecting multiple biopsies from different tumor regions, evaluating multiple histologic sections per biopsy, and integrating immunohistochemical results across sections to capture intratumoral variability. Additionally, incorporating digital pathology tools and quantitative image analysis may enhance the consistency and reproducibility of PD-L1 scoring. Future studies should aim to standardize biopsy protocols and validate these approaches to ensure robust biomarker assessment in clinical practice) (line 175-187). 

Comment 6. The role of digital pathology or AI in standardizing PD-L1 assessment is not mentioned, despite recent studies supporting its usefulness.

Response: Thank you for your insightful comment. In response, we have added a new subsection summarizing recent studies on the use of digital image analysis and artificial intelligence in PD-L1 CPS scoring. This addition highlights their potential to enhance the standardization and reproducibility of PD-L1 assessment (line 260-271).

 Recommendations

a) Clarify the methodological design and, if applicable, include a “Materials and Methods” section detailing data collection, inclusion criteria, and statistical analysis.

b) Include a balanced discussion of the limitations of interchangeability, especially in contexts of heterogeneous expression or near the cutoff point.

c) Include a discussion on the need for external validation in populations with lower incidence of gastric cancer.

d) Include a brief guide or reference to international standards (e.g., IASLC, CAP) for the validation of LDTs in PD-L1.

e) Include practical recommendations based on recent evidence to improve the representativeness of biopsies.

f) Include a brief section or mention of the future of PD-L1 assessment with digital tools.

Response: We sincerely appreciate the reviewer’s thoughtful and detailed recommendations. Most of these points overlap with the specific comments addressed above, and corresponding revisions have been incorporated throughout the manuscript.

We appreciate the thoroughness of the reviewer and hope that these changes adequately address their concerns.

Reviewer 2 Report

Comments and Suggestions for Authors

Review Comments

       This manuscript shared the experiences from the authors together with learnings from the literature to provide practical guidance on how to improve consistency and reliability of PD-L1 assessment in gastric cancer samples, which will arouse the interest from the readers in the relevant field. Overall, this review article can be published in Biomedicines after some minor revisions, which can be concluded as follow:

  1. The section heading name “1. Introduction” should be supplemented before Line 33, and the section number of this manuscript should be reorganized.
  2. The summary content of the whole review article should be further enriched (Line 80-82).
  3. It is not necessary to make the text “pre-treatment process” bold in Line 90.
  4. Several minor errors: 1) The extra space between the words “of” and “diagnosed” in Line 73 should be removed; 2) a comma should be added after the phrase “In this paper” in Line 80; 3) the first letter of the word “cytoplasmic” in Line 106 should be capitalized.

Author Response

Dear Reviewer #2

The revised manuscript titled “Best Practice PD-L1 Staining and Interpretation in Gastric Cancer using PD-L1 IHC PharmDx 22C3 and PD-L1 IHC PharmDx 28-8 Assays, with Reference to Common Issues and Solutions” is enclosed for consideration as a review article in Biomedicines. Our responses are detailed below. In the revised manuscript, the changes are highlighted in red.

This manuscript shared the experiences from the authors together with learnings from the literature to provide practical guidance on how to improve consistency and reliability of PD-L1 assessment in gastric cancer samples, which will arouse the interest from the readers in the relevant field. Overall, this review article can be published in Biomedicines after some minor revisions, which can be concluded as follow:

Comment 1. The section heading name “1. Introduction” should be supplemented before Line 33, and the section number of this manuscript should be reorganized.

Response: Thank you for your helpful comment. In response, we have inserted the section heading “1. Introduction” before line 37 and have renumbered all subsequent sections sequentially to ensure consistency throughout the manuscript (highlighted in red).

Comment 2. The summary content of the whole review article should be further enriched (Line 80-82).

Response: Thank you for your valuable suggestion. In response, we have enriched the Summary section to more effectively emphasize the key findings and messages of the review (line 263-271 in the final manuscript).

 Comment 3. It is not necessary to make the text “pre-treatment process” bold in Line 90.

Response: Thank you for pointing this out. We have removed the bold formatting of “pre-treatment process” to maintain consistency with the formatting used in other sections (line 94).

 Comment 4. Several minor errors: 1) The extra space between the words “of” and “diagnosed” in Line 73 should be removed; 2) a comma should be added after the phrase “In this paper” in Line 80; 3) the first letter of the word “cytoplasmic” in Line 106 should be capitalized.

Response: Thank you for pointing out these minor errors. We have corrected the extra spacing, added the appropriate punctuation, and capitalized the word as suggested. Additionally, we have reviewed the entire manuscript to ensure typographical consistency.

We appreciate the thoroughness of the reviewer and hope that these changes adequately address their concerns.

Reviewer 3 Report

Comments and Suggestions for Authors

The review addresses an important practical problem (PD-L1 CPS scoring and assay concordance in gastric cancer) and offers helpful laboratory troubleshooting tips and a concise literature summary. However, the manuscript lacks methodological transparency, quantitative detail of the authors’ own data, consistent presentation of statistics, and some formatting issues.

I have the following comments:

-The paper reads as an expert guidance/review but does not state how the literature was identified, what the inclusion/exclusion criteria were, or whether this is a systematic or narrative review. Please add a short Methods section specifying search strategy etc, or explicitly state that this is an expert narrative review and describe how evidence and examples were selected.

-The manuscript refers to the authors’ laboratory experience (e.g., “we assess >50 gastric cancer samples per week”) but gives no systematic data, time frame, or methods for how examples/non-specific staining frequency were derived. Lines ~76–81 and other sections refer to lab experience.  Please provide a concise methods/results subsection for the authors’ case series: number of cases analyzed, period, sample types (biopsies, resections), how non-specific staining was defined and scored, rates (%) of non-specific staining by assay, and any statistical comparisons. If detailed quantitative data are not available, explicitly state this and limit conclusions accordingly.

-The manuscript "jumps" from section 4 to section 6 (Summary). This is confusing and suggests a missing section or formatting error.  Please fix numbering and ensure all intended sections are present.

-Table 1 reports OPA, Kappa, Cohen’s, Gwet’s etc. but the manuscript does not explain differences in metrics or how to interpret conflicting values across studies (also sample type TMA vs resections differs). Please add a short paragraph explaining differences between metrics (OPA vs kappa vs Gwet’s vs ICC), state how different sample types (TMA, biopsy, resection) affect concordance, and — if possible —reframe comparisons using a consistent metric (or add CIs). At minimum, add commentary in the text and footnotes explaining comparability limitations.

-Several numerical claims (e.g., “non-specific staining in 29/55 cases”) are cited from literature but the authors’ own experience lacks numbers, confidence intervals, or statistical testing. Wherever numerical comparisons are made (including the authors’ own data), provide sample sizes, percentages, and confidence intervals. State statistical tests used when comparing assays or reporting agreement.

-The clinical consequences of assay discordance at specific CPS thresholds (CPS ≥1 and ≥5) need clearer discussion — e.g., how interchangeability affects treatment eligibility (pembrolizumab vs nivolumab) and resulting patient management. The manuscript states differing approvals (lines ~33–45) but does not explicitly map assay discordance to possible clinical misclassification.

-The authors suggest: ‘Place wet paper towels under the staining area’. This is practical but potentially non-standard or perhaps unsafe for equipment warranty. The statement “Autostainer Link 48 is not a perfectly closed system” needs evidence or a manufacturer reference.  Please replace or supplement informal tips with manufacturer-endorsed procedures (cite vendor manuals where appropriate), describe acceptable alternative technical solutions (e.g., humidity chambers, rack replacement schedule), and if non-standard measures are recommended, state them as local/temporary measures and discuss safety/QA implications.

-Digital pathology is mentioned as potentially useful (lines ~231–233) but the manuscript does not describe how it can be implemented, validated, or what evidence supports its use for CPS. Please expand this section.

-Provide one-paragraph boxed guidance with bullet points on what to exclude from CPS (e.g., cytoplasmic only staining, glands, muscle, neutrophils) and how to confirm with H&E. This helps readers implement the guidance.

-The recommendation to average CPS across fragments (lines ~142–147) should be accompanied by an explicit SOP suggestion (how many fragments to average, when to report a range vs mean). Provide a short flowchart.

-A few sentences are awkwardly phrased; a light copyedit is recommended (e.g., “The Autostainer Link 48 is not a perfectly closed system, leading to a vulnerability where it dries out easily” — rephrase for clarity).

-Consider adding a one-page “laboratory checklist” as a supplement: pre-analytic (fixation, slide storage), analytic (controls, rack inspection), post-analytic (reporting language, when to reflex), and training recommendations.

-Consider providing a recommended maximum age for archival blocks/slides or when re-cutting is necessary.

Comments on the Quality of English Language

editing for syntax and typos

Author Response

Dear Reviewer #3

The revised manuscript titled “Best Practice PD-L1 Staining and Interpretation in Gastric Cancer using PD-L1 IHC PharmDx 22C3 and PD-L1 IHC PharmDx 28-8 Assays, with Reference to Common Issues and Solutions” is enclosed for consideration as a review article in Biomedicines. Our responses are detailed below. In the revised manuscript, the changes are highlighted in red.

The review addresses an important practical problem (PD-L1 CPS scoring and assay concordance in gastric cancer) and offers helpful laboratory troubleshooting tips and a concise literature summary. However, the manuscript lacks methodological transparency, quantitative detail of the authors’ own data, consistent presentation of statistics, and some formatting issues. 

Comment 1. The paper reads as an expert guidance/review but does not state how the literature was identified, what the inclusion/exclusion criteria were, or whether this is a systematic or narrative review. Please add a short Methods section specifying search strategy etc, or explicitly state that this is an expert narrative review and describe how evidence and examples were selected.

Response: Thank you for your thoughtful comment. As clarified in the revised Introduction and Abstract, this article is a narrative review that synthesizes published evidence with the authors’ interpretive commentary and laboratory experience. It does not adhere to a systematic review protocol. We have now explicitly stated this and described the approach used to select and integrate relevant evidence (line 18-23).

Comment 2. The manuscript refers to the authors’ laboratory experience (e.g., “we assess >50 gastric cancer samples per week”) but gives no systematic data, time frame, or methods for how examples/non-specific staining frequency were derived. Lines ~76–81 and other sections refer to lab experience. Please provide a concise methods/results subsection for the authors’ case series: number of cases analyzed, period, sample types (biopsies, resections), how non-specific staining was defined and scored, rates (%) of non-specific staining by assay, and any statistical comparisons. If detailed quantitative data are not available, explicitly state this and limit conclusions accordingly.

Response: Thank you for highlighting this issue. We have removed the unpublished quantitative statement (“we assess >50 gastric cancer samples per week”) from the manuscript. The discussion regarding the frequency of non-specific staining and related laboratory observations is based on data from our previously published study. We have clarified this in the revised text and added the appropriate reference to indicate that these findings are derived from our own published work.

Comment 3. The manuscript "jumps" from section 4 to section 6 (Summary). This is confusing and suggests a missing section or formatting error. Please fix numbering and ensure all intended sections are present.

Response: Thank you for your careful observation. We have corrected the numbering error, and all section and subsection numbers have been revised to follow a logical and sequential order throughout the manuscript.

Comment 4. Table 1 reports OPA, Kappa, Cohen’s, Gwet’s etc. but the manuscript does not explain differences in metrics or how to interpret conflicting values across studies (also sample type TMA vs resections differs). Please add a short paragraph explaining differences between metrics (OPA vs kappa vs Gwet’s vs ICC), state how different sample types (TMA, biopsy, resection) affect concordance, and — if possible —reframe comparisons using a consistent metric (or add CIs). At minimum, add commentary in the text and footnotes explaining comparability limitations.

Response: Thank you for your insightful comment. As you correctly noted, studies have employed various concordance metrics, and the reported levels of agreement may vary depending on both the metric used and the sample type (e.g., TMA, biopsy, resection). To address this, we have added a brief explanatory paragraph in the manuscript outlining the differences between commonly used metrics (e.g., OPA, Cohen’s kappa, Gwet’s AC1, ICC) and how sample type may influence concordance. We have also included commentary in the main text and footnotes of Table 1 to clarify the limitations in comparing results across studies due to these methodological differences (line 224-225).

Comment 5. Several numerical claims (e.g., “non-specific staining in 29/55 cases”) are cited from literature but the authors’ own experience lacks numbers, confidence intervals, or statistical testing. Wherever numerical comparisons are made (including the authors’ own data), provide sample sizes, percentages, and confidence intervals. State statistical tests used when comparing assays or reporting agreement.

Response: Thank you for your thoughtful suggestion. In response, we have rephrased or removed statements that lacked supporting source data. Where applicable, we have ensured that numerical comparisons are accompanied by sample sizes, percentages, and references to published data. We have also clarified that statistical analyses were not performed on the authors’ own observations, and have limited conclusions accordingly.

 Comment 6. The clinical consequences of assay discordance at specific CPS thresholds (CPS ≥1 and ≥5) need clearer discussion — e.g., how interchangeability affects treatment eligibility (pembrolizumab vs nivolumab) and resulting patient management. The manuscript states differing approvals (lines ~33–45) but does not explicitly map assay discordance to possible clinical misclassification.

Response: Thank you for your insightful comment. We have expanded the discussion to clarify the clinical implications of assay discordance at CPS thresholds (≥1 and ≥5). Even minor discrepancies between assays can result in divergent treatment eligibility decisions. For instance, a patient classified as CPS 4 by the 28-8 assay but CPS 5 by the 22C3 assay would be eligible for pembrolizumab but not for nivolumab. Such discordance may lead to clinical misclassification and directly impact treatment selection and patient management (line 67-78).

Comment 7. The authors suggest: ‘Place wet paper towels under the staining area’. This is practical but potentially non-standard or perhaps unsafe for equipment warranty. The statement “Autostainer Link 48 is not a perfectly closed system” needs evidence or a manufacturer reference. Please replace or supplement informal tips with manufacturer-endorsed procedures (cite vendor manuals where appropriate), describe acceptable alternative technical solutions (e.g., humidity chambers, rack replacement schedule), and if non-standard measures are recommended, state them as local/temporary measures and discuss safety/QA implications.

Response: Thank you for this important point. We agree that the suggestion to place wet paper towels under the staining area represents a local, non-standard troubleshooting practice. In the revised text, we have clarified that this is a temporary measure not endorsed by the manufacturer, and we have emphasized that users should adhere to appropriate safety and quality assurance protocols. Where applicable, we have supplemented the discussion with manufacturer-recommended procedures and alternative technical solutions line 115-118).

Comment 8. Digital pathology is mentioned as potentially useful (lines ~231–233) but the manuscript does not describe how it can be implemented, validated, or what evidence supports its use for CPS. Please expand this section.Response: Thank you for your thoughtful suggestion. In response, we have added a new subsection that summarizes recent studies on the implementation and validation of digital image analysis and artificial intelligence for PD-L1 CPS scoring. This addition highlights the supporting evidence for their use and discusses their potential role in improving consistency and diagnostic accuracy (line 275-283).  Comment 9. Provide one-paragraph boxed guidance with bullet points on what to exclude from CPS (e.g., cytoplasmic only staining, glands, muscle, neutrophils) and how to confirm with H&E. This helps readers implement the guidance.

Response: Thank you for your helpful suggestion. In response, we have added a boxed summary outlining key elements to exclude from CPS evaluation—such as cytoplasmic-only staining, glands, muscle, and neutrophils—and how to confirm these findings using H&E staining. This guidance is based on the PD-L1 IHC 28-8 pharmDx Interpretation Manual for gastric cancer (line 167).

Exclusions and H&E Confirmation (adapted from PD-L1 IHC 28-8 pharmDx Manual for Gastric Cancer)

  • Exclude from CPS:
    • Tumor cells with cytoplasmic-only staining
    • PD-L1–positive benign glands or metaplastic epithelium
    • Fibroblasts, smooth muscle, endothelial cells
    • Neutrophils, eosinophils, plasma cells
    • Necrotic or mucinous debris
  • Confirm with H&E:
    • Verify PD-L1–positive cells correspond to viable tumor or mononuclear immune cells.
    • Reassess areas with ambiguous staining to avoid inclusion of benign or stromal elements.

Comment 10. The recommendation to average CPS across fragments (lines ~142–147) should be accompanied by an explicit SOP suggestion (how many fragments to average, when to report a range vs mean). Provide a short flowchart.

Response: Thank you for your constructive suggestion. While we acknowledge the value of including an explicit SOP and flowchart, we believe that a concise description sufficiently conveys the intended approach given the straightforward nature of the procedure. In cases of heterogeneous PD-L1 staining, the final CPS was calculated as the mean value across evaluable tissue fragments. We have added this clarification to the manuscript. Reporting a range is not routinely practiced in pathology, as pathologists are generally familiar with this evaluation method (line 172-173). 

Comment 11. A few sentences are awkwardly phrased; a light copyedit is recommended (e.g., “The Autostainer Link 48 is not a perfectly closed system, leading to a vulnerability where it dries out easily” — rephrase for clarity).

Response: Thank you for pointing this out. The sentence has been revised for clarity and accuracy (line 113-114):
“Although the Autostainer Link 48 is designed as a closed system, occasional drying of tissue sections can occur for reasons that are not fully understood.”
The entire manuscript has also been rechecked for clarity and consistency.  

Comment 12. Consider adding a one-page “laboratory checklist” as a supplement: pre-analytic (fixation, slide storage), analytic (controls, rack inspection), post-analytic (reporting language, when to reflex), and training recommendations.

Response: Thank you for your kind suggestion. In response, we have added a one-page “laboratory checklist” as Supplementary Table 1, covering pre-analytic, analytic, and post-analytic considerations, as well as training recommendations (line 139-140). 

Supplementary Table S1. PD-L1 Laboratory Checklist for Gastric Cancer

Step Checklist Item Key Points / Recommendations
1. Specimen adequacy (Pre-analytical) Minimum tumor content At least 100 viable tumor cells should be present for valid PD-L1 evaluation.
  Fixation and embedding Formalin-fixed, paraffin-embedded (FFPE) tissue with 6–48 h fixation in 10% neutral buffered formalin.
  Block age Use recent FFPE blocks (<3 years old when possible)
  Section storage Cut sections 4–5 µm; store at 2–8 °C and stain within manufacturer’s recommended period (e.g., ≤5 months for 22C3).
2. Analytical phase Assay selection Use FDA-approved companion diagnostic assays (e.g., PD-L1 IHC 22C3, 28-8)
  Platform verification Ensure platform compatibility (Autostainer Link 48, Benchmark, etc.) and validated staining protocol.
  Controls Include on-slide positive and negative control tissues for each batch.
3. Interpretation and reporting (Post-analytical) Scoring system Use Combined Positive Score (CPS = [PD-L1+ tumor + PD-L1+ immune cells]/total tumor cells ×100).
  Exclude from CPS count • Cytoplasmic-only tumor staining
• PD-L1+ normal glands/metaplasia
• Smooth muscle, fibroblasts, endothelial cells
• Neutrophils/plasma cells/ eosinophils
• Necrotic/mucinous debris
  Confirm with H&E Verify PD-L1+ cells correspond to viable tumor or mononuclear immune cells on H&E.
  Reporting details Specify assay clone, platform, cutoff (e.g., CPS ≥1, ≥5), sample type (biopsy/resection), and adequacy statement.
4. Quality assurance Internal QC Monitor inter-run and inter-observer concordance; document deviations.
  External QA Participate in external proficiency testing or EQA programs when available.
  Documentation Record reagent lot numbers, control performance, and review log for all cases.

Comment 13. Consider providing a recommended maximum age for archival blocks/slides or when re-cutting is necessary. Response: Thank you for your helpful suggestion. We were unable to identify a specific guideline regarding the maximum archival age of FFPE blocks in gastric cancer. However, a reference guideline for non–small cell lung cancer (IASLC Atlas of PD-L1 Immunohistochemistry Testing in Lung Cancer) recommends using FFPE blocks within approximately 3 years. We have added the following sentence to the manuscript (line 135-139):
“When using archival formalin-fixed paraffin-embedded (FFPE) blocks, it is recommended to select the most recent specimen whenever possible to minimize antigen degradation. According to one reference guideline for PD-L1 testing in non-small cell lung cancer, FFPE blocks should preferably be used within approximately three years”Ref. Tsao MS, Kerr KM, Dacic S, et al. IASLC Atlas of PD-L1 Immunohistochemistry Testing in Lung Cancer. Aurora (CO): IASLC; 2017. ISBN 978-0-9832958-7-7.  

We appreciate the thoroughness of the reviewer and hope that these changes adequately address their concerns.

Round 2

Reviewer 1 Report

Comments and Suggestions for Authors The authors' comments and/or contributions, in my opinion, justify their publication in this latest version. No objection. Congratulations, well done, and thank you very much.

Reviewer 3 Report

Comments and Suggestions for Authors

The authors greatly improved the manuscript.